Genetic diversity of indigenous guinea fowl (Numida meleagris) using microsatellite markers in northern Togo

Soara Aïcha Edith edith.soara@gmail.com 1 2
Talaki Essodina 1 3
Dayo Guiguigbaza-Kossigan 4
Houaga Isidore 5
Tona Kokou 1 3
Bakkali Mohammed 6
1 Centre d’Excellence Régional sur les Sciences Aviaires, Université de Lomé , Lomé , Togo
2 Département Environnement et Forêts, Institut de l’Environnement et de Recherches Agricoles , Ouagadougou , Burkina Faso
3 Ecole Supérieure d’Agronomie, Université de Lomé , Lomé , Togo
4 Institut du Sahel , Bamako , Mali
5 Centre International de Recherche-Développement sur l’Elevage en zone Subhumide , Bobo-Dioulasso , Burkina Faso
6 Departamento de Genética, Facultad de Ciencias, Fuentenueva, Universidad de Granada , Granada , Spain
Edwards Scott
Electronic publication date: 2022 Jan 20
Publication date: 2022
Volume: 10
Electronic Location ID: e12637
Received 2021 Jun 18; Accepted 2021 Nov 23
Copyright: ©2022 Soara et al.
Copyright year: 2022
Copyright holder: Soara et al.
License: This is an open access article distributed under the terms of the Creative Commons Attribution License, which permits unrestricted use, distribution, reproduction and adaptation in any medium and for any purpose provided that it is properly attributed. For attribution, the original author(s), title, publication source (PeerJ) and either DOI or URL of the article must be cited.
License URL: https://creativecommons.org/licenses/by/4.0/

Keywords: Indigenous guinea fowl, Genetic diversity, Microsatellite markers, Togo

Funding: The German Academic Exchange Service Deutscher Akademischer Austauschdienst (DAAD) through the In-Country/In-Region scholarship program The present study was financially supported by the German Academic Exchange Service Deutscher Akademischer Austauschdienst (DAAD) through the In-Country/In-Region scholarship program. The funders had no role in study design, data collection and analysis, decision to publish, or preparation of the manuscript.

==============================
Indigenous guinea fowl is an important animal resource for improving rural household income. In order to provide molecular data for a sustainable management of this poultry resource, an assessment of the genetic diversity and phylogenic relationships was undertaken on seven guinea fowl phenotypes from two agroecological zones (Dry Savannah and Atakora) of Togo. Genotyping was carried out using 18 microsatellite markers on 94 individuals from Dry Savannah (59) and Atakora (35) zones. The results obtained showed a high genetic diversity, with six as an average alleles per locus and an observed heterozygosity of 0.512. However, the FIS values varied from 0.047 (Lavender) to 0.257 (Albino), reflecting a deficit of heterozygotes, which suggests low to moderate inbreeding levels. The genetic distances between phenotypes are low, ranging from 0.0068 (Bonaparte-Pearl grey) to 0.1559 (Lavender-Albino), unlike the strong genetic identities that reflect a strong genetic similarity between the seven phenotypes of indigenous guinea fowl studied. These results indicate the existence of a single indigenous guinea fowl population, derived from three probable parental populations, with a high within population genetic diversity (phenotypic or agroecological zone). These results could be of use to conservation and improvement programs aiming at the maintenance and sustainable exploitation of this important socio-cultural and economic resource in Togo.

Introduction

Helmeted guinea fowl (Numida meleagris) belonging to the order Galliformes, domesticated in West-Africa, probably in Mali and Sudan (Vignal et al., 2019). Its production provides many services to farmers, particularly meat, eggs, serving as an important protein source for consumers and therefore play an indispensable role in food security for the people. Despite these products provided by indigenous guinea fowl, it is still conidered as a traditional village poultry. In Togo, indigenous guinea fowl farming is widespread, especially in the two northern agroecological zones where it is an important source of income for rural households (Lombo, Tona & Bonfoh, 2018). The production of this poultry species is mainly practised in semi-intensive systems in which the birds are kept in a poultry house and have free access to a pasture area during the day thus constituting a randomly mating unselected population that constitute a huge treasure of variable genotypes (Yeasmin, Howlider & Ahammad, 2003). Due to the harsh environmental conditions under which the indigenous guinea fowl is kept, they may contain important genes and alleles for their adaptation to particular environments (Osei-Amponsah et al., 2010). Recent awareness of the value of genetic resources has encouraged studies on the genetic diversity in domestic animal species. Studies were carried out on the genetic diversity and population structuring of indigenous guinea fowl in Sub-Saharan Africa (Kayang et al., 2010; Weimann et al., 2016; Traoré et al., 2018). However, no information is available on the genetic diversity and structure of indigenous guinea fowl in Togo despite their socio-economic importance. Phenotypic characterization showed a great diversity between guinea fowl of Dry Savannah and Atakora agroecological zones on the one hand and between the different phenotypes based on the plumage colours on the other hand (Soara et al., 2020). However, it did not allow the specific identification of breeds, strains or ecotypes. Yet, it is important to determine whether indigenous guinea fowl populations in these two agroecological zones of Togo represent unique populations for the development of effective conservation programs, as well as to assess their diversity at the molecular level in order to provide recommendations regarding their future management and conservation (Osei-Amponsah et al., 2010). The guinea fowl has four known loci that influence its plumage colour and all of these loci are autosomal with three being recessive and the fourth incompletely dominant. Colour varieties have been developed around these four mutations and their various combinations (Ghigi, 1924; Ghigi, 1966; Somes Jr, 1996). In the current study, our objective was to evaluate the genetic variability in each plumage colour phenotype identified by the phenotypic characterization of guinea fowl of Togo (Soara et al., 2020). The use of polymorphism of molecular markers such as microsatellite loci is a reliable way of assessing genetic diversity within and between populations (Hillel et al., 2003; Osei-Amponsah et al., 2010; FAO, 2011; Fotsa et al., 2011). Indeed, microsatellite loci are abundant, randomly distributed in the genome, highly polymorphic and show co-dominant inheritance (Hillel et al., 2003; Ocampo, Cardona & Martínez, 2016; Vieira et al., 2016). To date, only few specific markers have been developed for guinea fowl (Botchway et al., 2013). Fortunately, cross-species amplification of microsatellite loci has been reported within closely related avian species (Kayang et al., 2002; Nahashon et al., 2008). For instance, microsatellite markers developed in other avian species such as chicken and quail have been used to assess the genetic diversity of domestic and wild guinea fowl populations in Benin, Ghana, Sudan and Burkina Faso (Kayang et al., 2010; Weimann et al., 2016; Traoré et al., 2018). The present study takes advantage of the available microsatellite markers that have been proven to work in the guinea fowl species to characterized for the first time the indigenous guinea fowl populations of Togo. The aim of this study was to assess the genetic diversity and structure in the main colour plumage phenotypes and in the indigenous guinea fowl populations in two agroecological zones of the northern Togo. The information and data generated could be used for management, improvement and/or conservation strategies of guinea fowl in Togo.

Materials and Methods

Study areas

The study was conducted between March and July 2018 in the fourteen (14) prefectures of the two agroecological zones (Dry Savannah and Atakora) in Northern Togo (Fig. 1). With about 1,085,260 guinea fowl in 2012 (64% of national production), the Dry Savannah and Atakora are the main zones of guinea fowl production in Togo (FAO, 2015). The north of Togo is located between 0° and 1° East longitude and 9° and 11° North latitude. It is bounded to the North by the Republic of Burkina Faso, to the East by the Republic of Benin, to the West by the Republic of Ghana and to the South by the Wet Savannah zone.

Figure 1 Map of Dry Savannah and Atakora agroecological zones in northern Togo with the locations of guinea fowl sampled.

– The Dry Savannah zone, located in extreme North of Togo, is divided administratively into eight prefectures: Cinkasse, Tone, Tandjouare, Kpendjal, Western Kpendjal, Oti, South Oti, and Keran. It is a lowland area and covers an area of 10,573 km2. The human population, mainly rural, was estimated to be 985,900 inhabitants in 2013 (INSEED, 2019), with a growth rate of 3.18 percent. The climate of this zone is Sudanese, characterised by 1,000–1,100 mm of annual rainfall and 28.5 °C of annual average temperature. The average humidity level of the area is 56% (Amey et al., 2014). The main vegetation consists of savannah woodland dominated by Andropogon tectorum and Combretum molle and shrubby savannah dominated by Andropogon tectorum and Crotalaria graminicola (Demakou, 2009). The majority of the zone inhabitants are peasant and subsistent farmers.

– The Atakora zone includes six prefectures: Kozah, Bassar, Doufelgou, Dankpen, Binah and Assoli and covers about 9,742 km2. It is a mountains area with a sudano-guinean climate. The zone receives an average annual rainfall of 1,250 mm and a minimum and maximum temperature of 21 °C and 34 °C, respectively. The average humidity level is 63% (Amey et al., 2014). The natural vegetation is made of up of sudanian savannah, opened and dry forests (Nabede et al., 2018). The human population, estimated to be 711,500 inhabitants in 2013 (INSEED, 2019), is entirely agriculturists.

DNA sampling and extraction

A total of 94 adult guinea fowl were sampled in 26 villages belonging to the 14 prefectures of the two agroecological zones in northern Togo (Fig. 1). In each village, one to seven guinea fowl specimens were randomly selected. Thus, 59 birds were sampled from the Dry Savannah zone and 35 from the Atakora zone. Approximately 1.5 mL of whole blood was collected from the wing vein of each sampled bird using 21G vacutainer needles and 5 mL vacutainer tubes containing ethylenediaminetetra-acetic acid (EDTA) as anticoagulant. The tubes were transported under cold conditions to the laboratory and stored at +4 °C. DNA was extracted at the International Center for Research and Development on Livestock in the Subhumid zone (CIRDES/Centre International de Recherche-Développement sur l’Elevage en zone Subhumide) according the PIFLOTOU method. This method allows the extraction of DNA from nucleated red blood cells in birds. It consists of extracting DNA from 20 µL of whole blood in 600 µL of lysis solution (60 µL of Tris HCl + 120 µL of EDTA + 150 µL of NaCl + 250 µL of SDS + 20 µL of Proteinase K). After incubation step at 37 °C overnight, 250 µL of saturated NaCl (usually cold) was added per sample and the mixture was centrifuged for 30 min at 14,000 revolutions per minute. The majority of the supernatant was carefully transfered to a tube containing 2 mL of absolute ethanol (usually cold). The samples were let stand at +4 °C about 1 h to allow the DNA to fully precipitate. The clot of DNA fibres was collected and placed in 2 mL sterile tube containing 250 µL of Tris EDTA (TE) and incubated at 37 °C overnight. After dissolution, DNA was quantified by spectrophotometry. The purity and concentration of the extracted DNA were measured using a 1000 NanoDrop Spectrophotometer. The tubes containing the DNA were stored at −20 °C until the amplification and the genotyping assays.

Table 1 Information on multiplexes and amplification temperatures.

Multiplexes	Amplification temperature (°C)	Loci	Allele size (bp)	Fluorochrome	
Multiplex 1	55	GF43
GUJ0001
GUJ0059
GUJ0066	111–117
222–226
211–231
161–255	VIC
VIC
NED
PET	
Multiplex 2	55	GF13
GF5
GUJ0084
GF30
GF75	114–136
165–167
165–167
192–202
207–217	FAM
VIC
NED
NED
PET	
Multiplex 3	60	GF168
GF12
GUJ0013
GF37
GUJ0086
MCW0222	214–230
100–110
134–150
206–230
205–213
214–230	FAM
FAM
FAM
VIC
NED
PET	
Multiplex 4	60	GF74
GF69
MCW0069	208–214
182–200
182–200	FAM
VIC
PET	

PCR amplification and genotyping

A set of eighteen (18) microsatellite markers were chosen for this study: six microsatellite markers (GUJ0084, GUJ0013, GUJ0001, GUJ0066, GUJ0059 and GUJ0086) developed by Kayang et al. (2002) for guinea fowl, quails and hens; 10 microsatellite markers (GF75, GF12, GF43, GF5, GF74, GF69, GF13, GF168, GF37 and GF30) developed by Botchway et al. (2013) for guinea fowl and 02 microsatellite markers (MCW0069 and MCW0222) from the panel recommended by the FAO for the study of diversity in hens. The microsatellite markers were grouped in four optimised multiplexes (Table 1), and the 5′ end of the forward primer sequence of each microsatellite was accordingly labelled with one of the four fluorochromes (FAM, VIC, NED or PET). Multiplexed polymerase chain reactions (PCR) was performed in a final volume of 12 µL containing 10 ng DNA template, 20 pmol of each forward and reverse primer, 10x PCR buffer, 25 mM MgCl2, 25 mM dNTPs and 5 U Taq DNA polymerase. The amplification protocol involved an initial DNA denaturation step at 94 °C for 15 min followed by 30 cycles of denaturation at 94 °C (30 s), primer annealing at 55 °C or 60 °C (1 min) according to optimal temperature of the primers, extension at 72 °C (1 min) and a final extension at 72 °C for 15 min. An automated thermal cycler Veriti TM 96-Well Thermal cycler, Applied Biosystems was used. The amplified products generated by PCR were electrophoresed on an AB3500 automatic DNA sequencer using 8 capillaries (Applied Biosystems) at the CIRDES genotyping platform in Bobo-Dioulasso (Burkina Faso). The GeneMapper version 5.0 software (Applied Biosystems) was used to analyse the microsatellite markers electrophoretic profiles and estimate the size of the fragments (alleles at different loci) in base pairs.

Statistical analysis

The genotypes obtained were analysed in order to estimate the following polymorphism parameters: the total number of alleles, the average number of alleles per locus in all subpopulations, the allelic richness, the observed (Ho) and expected (He) heterozygosities. The factorial correspondence analysis (FCA), to visualise the differences between the studied subpopulations, and the Wright’s F-statistics FIS, FST and FIT (Wright, 1951) proposed by Weir & Cockerham (1984) were performed using GENETIX 4.05 (Belkhir et al., 2004) and FSTAT 2.9.4 (Goudet, 2005) software. The number of effective alleles and the polymorphism information content (PIC) for each locus were estimated using Molkin v.3.0 software (Gutiérrez et al., 2005). In addition, the unbiased measures of the pairwise genetic distance and genetic identity of Nei (1978) of the subpopulations were calculated using POPGENE version 1.32 (Yeh et al., 1999). In order to visualize the genetic relationships between phenotypes, the estimated distances were used to construct a phylogenetic tree following the Neighbour-Joining method using the PHYLIP software (Felsenstein, 1993). The genetic structuring of the population and the assignment of individuals to K populations were evaluated using the STRUCTURE 2.3.4 software (Pritchard et al., 2000). The distribution a posteriori of each individual’s probability values was inferred using a Bayesian approach. As genotyping information for the assumed parent populations was not available, we hypothesised K unknown populations of parents (K varying from two to six with 10 repeats for each K). The analysis was performed with a burn-in period of 50,000 iterations and 100,000 Markov Chain Monte Carlo (MCMC) based on the assumption of uncorrelated allele frequencies between parental populations and an admixture model. The optimal value of ‘K’ determining the most likely number of genetic clusters was identified on the basis of ΔK, the second order rate of change of LnP(D) proposed by Evanno, Regnaut & Goudet (2005) using the Structure Harvester programme (http://taylor0.biology.ucla.edu/structureHarvester/) (Earl & Vonholdt, 2012).

Results

Loci polymorphism and intra-population diversity

All the 18 microsatellite loci selected for genotyping the populations studied were polymorphic. A total of 108 alleles were identified from the 94 animals genotyped with an average number of 6.0 ± 5.1 alleles per locus (Table 2). The number of alleles per locus in the total population varied from two for loci GF5 and GUJ0084 to 25 for locus GUJ0066. The effective number of alleles per locus averaged 2.92 ± 1.06, ranging from 1.34 (GUJ0001) to 6.14 (GUJ0066). Across the agroecological zones, the average number of alleles was 5.2 alleles while this number varied across phenotypes, Black pied phenotype had the lowest average number of alleles (2.8) and Royal purple phenotype the highest (4.7 alleles) (Table 3). Within phenotypes, the allelic richness (AR) varied from 2.7 for the Black pied phenotype to 3.2 for the Pearl grey phenotype. The number of effective alleles ranged from 2.1 (Black pied) to 3.1 (Pearl grey). Compared to agroecological zones, the allelic richness was 4.9 and 4.7 for the Atakora and Dry Savannah subpopulations, respectively. The polymorphism information content (PIC) ranged from 0.235 (GUJ0001) to 0.825 (GUJ0066) with an average value of 0.556 ± 0.149. Out of the 18 loci studied, only six (GF43, GUJ0001, GF5, GUJ0084, GF30 and GF75) had a PIC value less than 0.5 (Table 2)

Table 2 Number of alleles (Na), allelic richness (AR), effective number of alleles (Ae), observed heterozygosity (Ho), expected heterozygosity (He), polymorphism information content (PIC) and Wright’s statistical F (FIT, FST and FIS) of the Atakora and Dry Savannah populations of indigenous guinea fowl in Togo.

Loci	Na	AR	Ae	Ho	He	PIC	F IT	F ST	F IS	
GF43	4	2.8	2.14	0.439	0.489	0.487	0.202	0.042	0.167	
GUJ0001	3	1.9	1.34	0.203	0.266	0.235	0.184	0.027	0.161	
GUJ0059	7	3.9	3.70	0.390	0.662	0.696	0.347	0.037	0.322	
GUJ0066	25	5.3	6.14	0.540	0.811	0.825	0.323	0.009	0.317	
GF13	7	3.6	3.40	0.502	0.655	0.663	0.258	0.026	0.238	
GF5	2	2.0	1.88	0.447	0.452	0.358	0.059	0.000	0.070	
GUJ0084	2	2.0	1.87	0.470	0.447	0.356	−0.014	0.001	−0.015	
GF30	3	2.1	1.97	0.308	0.454	0.386	0.329	0.022	0.314	
GF75	4	2.6	2.27	0.472	0.576	0.474	0.205	0.000	0.223	
GF168	4	3.2	3.17	0.663	0.703	0.621	0.064	0.000	0.088	
GF12	8	4.0	3.56	0.732	0.727	0.691	0.037	0.000	0.041	
GUJ0013	4	3.1	3.15	0.663	0.702	0.618	0.062	0.000	0.084	
GF37	7	3.6	3.61	0.816	0.740	0.673	−0.122	0.000	−0.106	
GUJ0086	5	3.0	2.92	0.548	0.651	0.589	0.227	0.000	0.229	
MCW0222	5	3.2	3.23	0.678	0.707	0.630	0.042	0.000	0.064	
GF69	7	3.2	2.84	0.502	0.644	0.592	0.277	0.015	0.266	
GF74	4	2.8	2.38	0.366	0.538	0.515	0.300	0.000	0.300	
MCW0069	7	3.3	2.92	0.487	0.646	0.599	0.301	0.031	0.279	
Average	6.00	3.08	2.92	0.512	0.604	0.556	0.171	0.012	0.169	
Standard deviation	5.10	0.85	1.06	0.154	0.137	0.149	0.139	0.015	0.128	

Across all subpopulations, the observed heterozygosity (Ho) ranged from 0.203 (GUJ0001) to 0.816 (GF37), with an average of 0.512 ± 0.154, while the expected heterozygosity (He) varied from 0.226 (GUJ0001) to 0.811 (GUJ0066), with an average of 0.604 ± 0.137. With the exception of GUJ0084, GF12 and GF37, all the other loci showed a heterozygote deficit compared to the expected according to the Hardy-Weinberg equilibrium (Ho < He) with FIS values significantly higher than zero (Table 2). The observed heterozygosity in the Dry Savannah guinea fowl subpopulation (0.522) was higher than the value obtained in the Atakora subpopulation (0.498). The Lavender phenotype had the highest values for observed (0.610) and expected (0.638) heterozygosity (Table 3). Apart from the Lavender and Black pied phenotypes, all the other phenotypes showed a significant deficit in heterozygotes. The FIS values suggest a significant heterozygote deficit in all but the Lavender and Black pied phenotypes.

Table 3 Average number of alleles (Na), allelic richness (AR), effective number of alleles (Ae), observed heterozygosity (Ho), expected heterozygosity (He) and deviation from panmixia (FIS) of the different populations.

Population	N	Na	AR	Ae	Ho	He	F IS	
Phenotypes								
Albino	5	3.2	3.0	2.6	0.447	0.584	0.257*	
Bonaparte	19	4.3	3.0	2.7	0.496	0.606	0.185*	
Pearl grey	19	4.6	3.2	3.1	0.521	0.633	0.181*	
Lavender	8	3.6	3.1	2.7	0.610	0.638	0.047	
Black pied	5	2.8	2.7	2.1	0.500	0.537	0.077	
Multi-coloured	18	4.3	3.0	2.7	0.499	0.616	0.196*	
Royal purple	20	4.7	3.1	2.8	0.514	0.617	0.170*	
Agroecological zones								
Atakora	35	5.17	4.9	3.24	0.498	0.619	0.199	
Dry Savannah	59	5.22	4.7	3.05	0.522	0.614	0.150	
Notes.

* p < 0.05.

Figure 2 The Factorial Correspondence Analysis (FCA) results.

(A) The relationship between phenotypes: Yellow (Albino), Blue (Bonaparte), White (Pearl grey), Grey (Lavender), Purple (Black pied), Green (Multi-coloured), Blue-black (Royal Purple). (B) The relations between guinea fowl populations of Atakora and Dry Savannah agroecological zones.

Inter-population diversity

Factorial Correspondence Analysis (FCA) showed a high degree of genetic similarity between the seven guinea fowl phenotypes studied (Fig. 2A) and between the two subpopulations of the agroecological zones studied (Fig. 2B).

The overall deficit of heterozygotes in the total population is reflected by FIT values ranging from −0.122 (GF37) to 0.347 (GUJ0059) with an average value of 0.171. The pairwise genetic distances between the seven indigenous guinea fowl subpopulations ranged from 0.0068 (Bonaparte-Pearl grey) to 0.1559 (Albino-Lavender). These genetic distances were inversely proportional to the genetic identities between pairs of subpopulations (Table 4). Between the two agroecological zones, the genetic distance and genetic identity were 0.0266 and 0.9737, respectively. The phylogenetic tree obtained from pairwise genetic distance of Nei (1978) between subpopulations is presented in Fig. 3. It shows a clade that groups the Pearl grey, Albino, Bonaparte, Black pied and Royal purple phenotypes, while the Lavender and the Multi-coloured phenotypes appeared in separated branches.

The most consistent gain in information was obtained with a number of clusters K = 3 (Fig. 4). Among the three groups defined at K = 3, Cluster 1 consisted mostly (39%) of individuals from the Dry Savannah, Cluster 2 predominated individuals from Atakora (43%) and Cluster 3 included 27% and 24% of individuals from the Dry Savannah and Atakora, respectively. This structuring shows the same trend as the FCA results.

Discussion

The aim of this study was to assess the diversity and genetic structure of the indigenous guinea fowl population in northern Togo using seven phenotypes from two agroecological zones as subpopulations. The present study is the first investigation on indigenous guinea fowl population in Togo using molecular markers. All loci were polymorphic with an average PIC value of 0.556, considered to reflect reasonably informative loci (Botstein et al., 1980). Indeed, based on the classification suggested by these authors, 66.7% (12/18) of the microsatellite markers used in this work were highly informative (PIC > 0.50), 27.8% (5/18) were reasonably informative (0.25 < PIC < 0.50) and 5.5% (1/18) slightly informative (PIC < 0.25). The use of a mixture of highly variable and less variable microsatellites should reduce the risk of overestimating genetic variability (Wimmers et al., 2000). The panel of 18 microsatellite markers used in the present study is therefore suitable for a genetic evaluation of the indigenous guinea fowl populations in Togo.

Table 4 Unbiased measures of genetic identity (above matrix) and genetic distances (below matrix) according to Nei (1978) between the seven phenotypes of indigenous guinea fowl in Togo (The high values are in Bold and the small values in italics and underlined).

	Albino	Bonaparte	Pearl grey	Lavender	Black pied	Multi-coloured	Royal purple	
Albino	−	0.9659	0.9492	0.8557	0.9244	0.9395	0.9271	
Bonaparte	0.0346	−	0.9932	0.8903	0.9581	0.9805	0.9736	
Pearl grey	0.0521	0.0068	−	0.9109	0.9464	0.9670	0.9718	
Lavender	0.1559	0.1162	0.0933	−	0.8566	0.9125	0.8688	
Black pied	0.0786	0.0428	0.0550	0.1548	−	0.9565	0.9492	
Multi-coloured	0.0624	0.0197	0.0335	0.0916	0.0445	−	0.9492	
Royal purple	0.0757	0.0267	0.0286	0.1406	0.0536	0.0522	−	

Figure 3 Phylogenetic tree representing Nei’ s genetic distance (Nei, 1978) between the seven phenotypes of indigenous guinea fowl in Togo.

Figure 4 Clustering diagram based on STRUCTURE analysis of the two agroecological zones (Atakora, Dry Savannah for K = 3).

Each line representing a single individual and the shading representing the three population clusters (Cluster 1: orange; Cluster 2: grey; Cluster 3: blue). The values on the x-axis represent the percentages of individuals from each agroecological zone in the different clusters.

A total of 108 alleles were found for the 18 microsatellite loci studied, with an average number of 6.0 alleles per locus. This value is similar to the findings of Traoré et al. (2018) in guinea fowl populations in Burkina Faso, but lower than the number reported by Kayang et al. (2010) in guinea fowl populations in Benin and Ghana (11 alleles per locus) and by Weimann et al. (2016) in wild and domestic guinea fowl in Sudan (9.7 alleles per locus). GUJ0066 locus showed the highest polymorphism in our study (25 alleles) as well as in those of Kayang et al. (2010) and Weimann et al. (2016) with 28 alleles and 36 alleles, respectively. In addition to the historic, environmental and biological differences between the studied populations, the differences observed between these studies could also be explained by the microsatellite markers used for genotyping and by the size of the studied samples. Indeed, the nature of loci used influences the average number of alleles per locus, and the number of alleles observed in a given locus in a population tends to increase with the size of the sample examined, which means that this parameter must be taken into account in comparisons of genetic diversity between populations (Ollivier & Foulley, 2013). To avoid these biases, correction methods (rarefaction and extrapolation) have been proposed (El Mousadik & Petit, 1996; Foulley & Ollivier, 2006). The estimation of allelic richness (AR) in our study is based on the rarefaction correction method implemented in FSTAT sofware. In the set of loci used in our work, the observed (Ho) and expected (He) heterozygosity of 0.512 and 0.604, respectively, were high. The Albino phenotype showed the lowest heterozygosities (Ho = 0.447 and He = 0.584) values while the Lavender phenotype showed the highest values (Ho = 0.610 and He = 0.638) (Table 3). Still, He was higher than Ho in all the subpopulations. Similar results were found by Kayang et al. (2010), Traoré et al. (2018) and Weimann et al. (2016) in guinea fowl populations of West Africa and Sudan, respectively. The higher overall rates of heterozygosity indicate a high genetic diversity, which is consistent with the great phenotypic variability observed previously in the same subpopulations (Soara et al., 2020). The great genetic diversity in indigenous guinea fowl in northern Togo could be the result of the absence of selection for specific phenotype or production traits and/or uncontrolled mating in the semi-intensive rearing system, which has as consequence a continuous genes flow between populations, the conservation of a high number of alleles, and a strong heterozygosity in the populations. However, an observed heterozygosity lower than that expected under the Hardy–Weinberg equilibrium hypothesis reflects a deficit of heterozygotes in the population that could be related to inbreeding or technical problems of amplification generating the null alleles. All the subpopulations studied here showed heterozygote deficit as also depicted by positive FIS value. The Wright’s fixation indices (FIS = 0.169, FST = 0.012 and FIT = 0.171) found in indigenous guinea fowl population of Togo were low. Similar results were reported in indigenous guinea fowl from Benin, Burkina Faso, Ghana and Sudan (Kayang et al., 2010; Weimann et al., 2016; Traoré et al., 2018). The low FST value (0.012) means that only 1.2% of the total genetic variability could be ascribed to differences between subpopulations, suggesting a low sub-structuring of the studied population. The low difference of the genetic variability between the studied phenotypes is corroborated by the low genetic distances and the high values of genetic identity between phenotypes and between agroecological zones.

Low values of genetic distances were also reported by Kayang et al. (2010) and Weimann et al. (2016). A genetic diversity study of India guinea fowl using Random amplified polymorphic DNA (RAPD) showed a very high and almost identical genetic similarity (0.97–0.99) between Lavender, Pearl grey and White guinea fowl phenotypes (Sharma et al., 1998). Bawej, Kokoszynski & Bernacki (2012) reported the same observation between the White and the Pearl grey guinea fowl in Poland. In our study, over the seven phenotypes, only the Lavender phenotype stood out slightly from the others. These results indicates that there is a weak genetic variation between the phenotypes (based on the feathers —plumage—colour) and between the two agroecological zones studied. The observed low genetic differentiation could be attributed to the many years of non-selective breeding (on the basis of feather colour) and to the uncontrolled movements of birds between agroecological zones and between border countries, which promotes gene flow (Sharma et al., 1998; Bawej, Kokoszynski & Bernacki, 2012; Weimann et al., 2016; Traoré et al., 2018). In addition, the microsatellite markers used in the present study are neutral markers and, therefore, are not involved in the determinism of the expression of feather colouring genes. The Atakora and Dry Savannah guinea fowl subpopulations are also genetically close (genetic identity of 0.9737) which could explain the low predominance observed in the three clusters identified by the a posteriori grouping from the STRUCRURE software. It also appears that the indigenous guinea fowl in northern Togo seem to come from three parental populations with some individuals specific to the Dry savannah zone or to the Atakora zone and others with a very heterogeneous genetic structure. It would be interesting to include in future studies, subpopulations from other agroecological zones of Togo but also guinea fowl populations from border countries such as Burkina Faso, Benin and Ghana. It would be also interesting to investigate on the genes underlying colour variation, most notably EDNRB2, in order to understand how these genes can affect a phenotype. EDNRB2 gene was selected as the most likely candidate gene due to its functional importance in melanocyte development and would explain various phenotypes found in domesticated animals (Vignal et al., 2019).

The results obtained in the current study could serve as baseline information to breeding strategies for the improvement and conservation on domestic guinea fowl populations in Togo. Because of the low or a lack of clear differentiation between guinea fowl population in the two agroecological zones, the future improvement strategy or program will not need to consider the different ecotypes. It is necessary that efforts will be made to improve the farming conditions such as the control of habitat, diet and diseases. Any strategy or program needs also to take into account the local community preferences, the different uses of the species (consumption, income generation, socio-cultural roles,…).

Conclusion

Traditionally, indigenous guinea fowl classification is based on the colour of the plumage. The present study showed a low genetic structuring in the seven phenotypes of indigenous guinea fowl in northern Togo. It also showed a heterozygotes deficit in the overall population and in the subpopulations represented by the phenotypes. The traditional rearing system, which is particularly based on wandering, allows gene flow between subpopulations of the indigenous guinea fowl population of Togo and no specific isolated genetic group was identified among the phenotypes studied. Despite a low level of differentiation, the high heterozygosity rates obtained show that there is an important reservoir of genetic diversity in the indigenous guinea fowl of Togo. This study provides a good genetic basis for establishing genetic improvement and conservation strategies for guinea fowl in Togo. The information generated here is useful for setting up programs of selection, genetic improvement and adaptation to tropical environmental conditions in the study area, as well as for a sustainable use of the indigenous guinea fowl of Togo. Such strategies and programs should also involve the indigenous communities in order to exploit their interesting endogenous knowledge and combine them with scientific knowledge for a better exploitation and conservation of the valuable economic and food resource that the guinea fowl is to the people of the studied area.

Supplemental Information

Supplemental Information 1 Genotypes by phenotypes

Click here for additional data file.

Supplemental Information 2 Genotypes by agroecological zones

Click here for additional data file.

We express our gratitude to the Regional Excellence Center on Poultry Sciences (CERSA) of University of Lome (Togo) and the World Bank Group. We also thank the staff of International Center for Research and Development on Livestock in the Subhumid zone (CIRDES) in Bobo-Dioulasso (Burkina Faso). This work also benefitted from the aid and help of the ERASMUS+ and CICODE programme and the University of Granada (Spain).

Additional Information and Declarations

Competing Interests

Author Contributions

Data Availability

The authors declare there are no competing interests.

Aïcha Edith Soara and Guiguigbaza-Kossigan Dayo conceived and designed the experiments, performed the experiments, analyzed the data, prepared figures and/or tables, authored or reviewed drafts of the paper, and approved the final draft.

Essodina Talaki conceived and designed the experiments, performed the experiments, prepared figures and/or tables, authored or reviewed drafts of the paper, and approved the final draft.

Isidore Houaga performed the experiments, authored or reviewed drafts of the paper, and approved the final draft.

Kokou Tona conceived and designed the experiments, authored or reviewed drafts of the paper, and approved the final draft.

Mohammed Bakkali performed the experiments, analyzed the data, authored or reviewed drafts of the paper, and approved the final draft.

The following information was supplied regarding data availability:

The raw data are available in Supplementary Files.

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
