# Peer review of "Genetic diversity of indigenous guinea fowl (Numida meleagris) using microsatellite markers in northern Togo"

_PeerJ, doi:10.7717/peerj.12637_

## Round 0.1 · original submission · Major Revisions

Reviewer 1 felt the study was well-formulated and robust, but reviewer 2 identified several shortcomings. In particular you may want to expand the introduction to set up the study in s more informative way. It is also not completely true that microsatellites are neutral since many sequence repeats are involved in gene regulation. You may want to cite a review by Brumfield, et al which discusses the relative merits of SNPs versus microsatellites in population genetics (Brumfield, R. et al. (2002). The utility of single nucleotide polymorphisms in inferences of population history. Trends in Ecology and Evolution. 18: 249-256)

·

Basic reporting

No comment.

Experimental design

No comment.

Validity of the findings

No comment.

Additional comments

Abstract
Line 27: ‘Indigenous guinea fowl are an important’. Please change ‘are’ to ‘is’.
Line 28: Please change ‘incomes’ to ‘income’.
Line 30: Please indicate the two agroecological zones.
Line 31: Why the use of 18 markers only? I thought the standard recommended by FAO is 25 and above. How were the 94 individuals sampled?
Line 38: How can you justify this assertion ‘three probable parental populations’?
Line 40: Please reconstruct the sentence.

Introduction
Line 51: Be cautious with the use of ‘is’ and ‘are’ in describing guinea fowl. If you are referring to a particular type or breed, use ‘is’. However, if you are making reference to more than one type or breed, use ‘are’.
Line 57-59: What about the use of mtDNA for maternal relationship?

Materials and Methods
Line 77-87: Kindly provide more information such as existing population of birds in the two zones, the vegetation and predominant agricultural activities and tribes.
Line 91-92: Why the difference in the number of samples for each zone?
Line 96: is this correct ‘International Center Research and Development on Livestock in Subhumid zone)?
Line 98: What standard range values did you use for this ‘purity and concentration’?
Line 102: Why only 18 markers?
Line 119: What did you use to view the bands before sequencing?
Line 134-135: You should have compared you NJ tree with that of maximum likelihood or maximum parsimony.


Results
Line 166: Change ‘Excepted from ‘ With the exception of’
NB: The result section is very good.

Discussion
Line 198: The study should have been complemented with mtDNA for a balanced report.
Line 201: Put a comma after’ (5/18)’
Line 208-211: Please try and discuss your findings before comparisons.
Line 235: What about birds in the wild? How have they influenced this diversity?
Lines 253 and 264: Why repeating the figures which have already been presented under ‘RESULTS’?

Conclusion
Line 273: Please rephrase.
Line 274: Please check

References
The references are adequate, but no consistency in the formatting.

Reviewer 2 ·

Basic reporting

No comment

Experimental design

No comment

Validity of the findings

No comment

Additional comments

Numida meleagris represents an important source of income for families living from agriculture in Africa. In the Introduction section, the authors begin justifying the study by the economic importance of the species in agriculture and also by the lack of information about the species' indigenous populations in Togo. Previous studies on genetic diversity based on analysis of mitochondrial DNA and microsatellite markers (Kayang et al. 2010; Botchway et al. 2013) revealed the absence of genetic structure in domesticated populations of Numida meleagris, indicating a recent domestication and rapid dispersal by Africa. Therefore, the study in natural populations is important because it represents a source of variability for the genetic improvement of the species. Although the aim of the authors is to assess the genetic diversity and structure of populations in northeastern Togo, they do not specify how the results will contribute to the management and improvement of conservation and management techniques. This gap stems from the lack of a hypothesis to be tested. In my view it could not be formulated since the authors do not report what is already genetically known about the domestication of the indigenous populations. The introduction lacks information. I suggest reading Vignal et al. (2019) that, although using a different molecular tool, the SNPs, draws a rich panorama regarding the domestication of the species in different regions of Africa. According to Vignal et al. (2019) there is an absence of genetic structure among domestic populations, however, there is divergence between these and native populations.

Another aspect that can only be noticed when reading the methodology is that the authors want to verify the possible relationship between genetic structuring and plumage color. In the introduction there is no mention of the mode of inheritance already defined by Ghigi (1924, 1966) for the guinea fowl. Different modes of inheritance, autosomal dominant or incomplete dominance, are involved and again the lack of basic genetic information on the subject makes it impossible to formulate a hypothesis to be tested regarding a possible correspondence between genetic structuring and plumage color.

Regarding the results, the number of base pairs of the amplicons of each locus should preferably be informed in Table 1. The compliance with the Hardy-Weinberg equilibrium should be tested for each locus and the results showed.

The authors said that increasing the sample number per phenotype concludes that it is necessary to include a greater number of individuals to be able to attribute their origin. One thing, in my view, is the data obtained with the sample number used to suggest an assignment. Another is that there is no attribution suggestion with the used markers. What is the sample number sufficient to make the analyzes robust? This evaluation must be prior to the study, establishing at least the minimum necessary sample size.

Lines 258-260, the authors conclude “In addition, the microsatellite markers used in the present study are neutral markers and therefore, are not involved in the determinism of the expression of feather coloring genes”. Therefore, if they had gone a little further into the introduction and included this already well-known evolutionary property of microsatellite markers, they could have predicted what would happen at the end of the study. For this purpose, some candidate gene locus, most notably EDNRB2 (Vignal et al. 2019), seem to be more appropriate. But the authors apparently contradict each other when on line 269 they consider that: “Increasing the sample size per phenotype would no doubt lead to more robust conclusions”. I mean, should the sample number or molecular marker be revised?

Regarding the replication of the methods, there is no reference to the DNA extraction method. The acronym mentioned is cited in the Thesis by Romé (2015) but also without a description of the methodology and was not found in another reference.

Still regarding the representation of the results, all of them are not self-explanatory. It is not informed which results are presented, regarding which species, from which location(s). On the map, for example, there is only the region of Togo, without locating it in Africa. In Weimann et al. (2016) is a good example, where the authors include the African continent in the image and highlight the studied location. For example: “Map of Togo in Africa with the locations of the X populations of guinea fowl genetically, Numida melagris. Authors just sated in the legend: Figure 1. Sampled sites locations. Of what, where, what was sampled? Figure 3 Agroecological zones – in the figure Atakora and “Dry” there is no mention of the region, the objective of the analysis, figure 5 there is no mention to the colors of the STRUCTURE diagram. All subtitles must be rewritten.

Of minor concern

Lines 271/273 change "structuration" to "structuring"

In citation of two authors change “and” to “&”

In references, adapt as the instructions to authors, eg, journal name in italics, no indent from second line onwards, remove the period from some references, no dot in authors' initials.

---

## Round 0.2 · Minor Revisions

The reviewers both feel the manuscript is substantially improved. Reveiwer 2 only has a small number of minor comments to be addressed.

·

Basic reporting

no comment

Experimental design

no comment

Validity of the findings

no comment

Additional comments

no comment

Reviewer 2 ·

Basic reporting

The authors were carefully dedicated to observing the suggestions submitted and were clear about the aspects that cannot be answered due to the prioritized experimental design. With the modifications made, the manuscript is better structured. Starting with the most consistent introduction, in terms of the presentation of the state of the art regarding studies in Numida meleagridis, relevant to the focus of the manuscript, for example, the objectives were defined, the mode of inheritance of plumage coloration was informed, the extraction methodology of the DNA, the size data of the amplified alleles.

Importantly, the inclusion made by the authors regarding the fact that plumage color can be an indicative aspect of genetic structuring. Therefore, the theme is more robust based on the bibliographical references added.

Experimental design

It still needs correction in Methods:
L. 153 It consists of "extracting" DNA from....
L. 155 After incubation.... 250 uL of saturated NaCl (usually cold) was added per sample.
Keep correcting all the description of the DNA extraction method using past tense to match the rest of the described methodology, once it has already been executed.
L. 173 rephrase the sentence. "reactions (PCR) was carried our using."

The data about Hardy-Weinberga expectations and the size of amplified alleles were also included.

L338-342 - It was important that the authors included a future perspective and the need for additional studies regarding the EDNRB2 gene as a likely candidate to investigate the occurrence of various phenotypes in domesticated populations.

Validity of the findings

No comment.

Additional comments

No more comments.

---

## Round 0.3 · accepted · Accept

Thank you for your revisions. You have addressed all of the previous reviewers' comments, and for that reason I am happy to recommend acceptance of your paper to PeerJ. Congratulations!